# No such thing as bad publicity? A quantitative content analysis of print media representations of primary care out-of-hours services

Hamish Foster,[1] Sara Macdonald,[1] Chris Patterson,[2] Catherine A O'Donnell[1]

¹General Practice and Primary Care, Institute of Health & Wellbeing, University of Glasgow, Glasgow, UK
²MRC/CSO Social & Public Health Sciences Unit, University of Glasgow, Glasgow, UK

**Correspondence to**
Professor Catherine A O'Donnell;
Kate.O'Donnell@glasgow.ac.uk

## ABSTRACT

**Objective** To explore how out-of-hours primary healthcare services (OOHS) are represented in UK national newspapers, focusing on content and tone of reporting and the use of personal narratives to frame stories.

**Design** A retrospective cross-sectional quantitative content analysis of articles published in 2005, 2010 and 2015.

**Data sources** Nexis database used to search 10 UK national newspapers covering quality, middle-market and tabloid publications.

**Inclusion/exclusion criteria** All articles containing the terms 'out-of-hours' (≥3 mentions per article) or ('NHS 24' OR 'NHS 111' OR 'NHS Direct') AND 'out-of-hours' (≥1 mention per article) were included. Letters, duplicate news items, opinion pieces and articles without a substantial portion of the story (>50% of an article's word count, as judged by researchers) concerning OOHS were excluded.

**Results** 332 newspaper articles were identified: 113 in 2005 (34.1%), 140 in 2010 (42.2%) and 79 in 2015 (23.8%). Of these, 195 (58.7%) were in quality newspapers, 99 (29.8%) in middle-market and 38 (11.3%) in tabloids. The most commonly reported themes were OOHS organisation, personal narratives and telephone triage. Stories about service-level crises and personal tragedy, including unsafe doctors and missed or delayed identification of rare conditions, predominated. The majority of articles (252, 75.9%) were negative in tone. This was observed for all included newspapers and by publication genre; middle-market newspapers had the highest percentage of negative articles (Pearson $\chi^2$=35.72, p<0.001). Articles presented little supporting contextual information, such as call rates per annum, or advice on how to access OOHS.

**Conclusion** In this first reported analysis of UK national newspaper coverage of OOHS, media representation is generally negative in tone, with frequent reports of 'negative exemplars' of OOHS crises and fatal individual patient cases with little or no contextualisation. We present recommendations for the future reporting of OOHS, which could apply to the reporting of healthcare services more generally.

## INTRODUCTION

Primary care out-of-hours services (OOHS) is defined as primary care provided when

### Strengths and limitations of this study

- ► Covers a comprehensive range of UK print media, including a representative spread of main publication genres, political leaning and readership demographics.
- ► Timespan covered by the analysis reflects a time of substantial change to the organisation of out-of-hours services in the UK.
- ► Only newspaper articles were included.
- ► Other popular news sources, for example, online news and social media, were excluded from this study.
- ► Analysis method may limit generalisability and researcher bias could not be eliminated.

family doctors' surgeries are closed (weekdays from 1800 hours to 0800 hours, weekends and public holidays) and is often delivered in community-based clinic settings. Such services generally provide care for conditions that are not life-threatening, in contrast to care delivered by hospital-based emergency departments, which is available at all times and designed to manage more urgent and life-threatening problems. OOHS have seen extensive policy and structural change in many healthcare systems.[1–4] In many European countries, this change was characterised by a switch from small rota-based systems of primary care physicians (general practitioners (GPs)) to regional cooperatives charged with OOHS responsibility.[5–7] In the UK, significant contractual change in 2004 offered GPs the option of transferring responsibility for OOHS to regional health organisations, leading to new models of care including out-of-hours centres, walk-in centres and minor injuries units, as well as the implementation of national or centralised telephone triage and advice services.[1 3 8 9] Difficulties in recruiting family doctors to work in the out-of-hours

period are one driver of these policy and structural changes.[4 7] A national audit of OOHS in England in 2014 found 60% of providers had gaps in their GP rotas.[10] In Scotland, older GPs typically contribute a disproportionate number of OOHS duty sessions, causing concern for future staffing of services.[11]

A second driver for change is increasing patient demand, due in part to ageing populations and associated multimorbidity,[12 13] and a concern about the supposed 'inappropriate' use of OOHS.[14 15] For example, a study of all out-of-hours calls to one region of Denmark between 2010 and 2011 found 24% of all out-of-hours calls were retriaged as 'medically inappropriate' and could have been redirected to in-hours services.[16] Difficulties accessing daytime primary care have also been reported as leading to increased use of OOHS.[17 18] Patients have also reported uncertainty about the urgency of presenting complaints, limited knowledge about when and how to access OOHS and confusion about which services to access, all of which may drive them towards using more visible and accessible emergency departments, rather than primary care OOHS.[19 20] Thus, patients need to be able to access information that helps them make decisions about where, and how, to access healthcare, especially for more urgent issues.

Print media remains a common source of public knowledge and a potentially powerful influence on peoples' perceptions. Newspapers are able to set public agendas, determining *what* issues are of public importance.[21] Newspapers also 'frame' issues, influencing *how* topics are viewed.[22] Negative newspaper coverage of an issue is not only associated with negative perceptions,[23] but with negative health behaviour outcomes, such as lower vaccine uptake.[24] Equally concerning are the numerous empirical examples of the misrepresentation of health issues by print media. For example, newspaper articles have been found to be overly positive in the reporting of new surgical interventions, to under-report epidemiological data in relation to the human papillomavirus vaccine[25 26] and to report stories about rare diseases more frequently than common conditions.[27] Media analyses in the UK have also suggested that negative newspaper portrayals of GPs, and GP pay, are linked to decreasing professional morale and GP recruitment difficulties.[28 29] However, to our knowledge, there has been no previous examination of how UK or international newspapers portray OOHS despite the aforementioned significant policy and structural changes. Improved understanding of this important influence on public, and staff, perceptions may inform OOHS service providers in their patient education campaigns and in staff recruitment drives.

This study aimed to explore how UK OOHS are portrayed in national newspapers, with a particular focus on the content and tone of reporting and the use of personal narratives or individual case studies to frame stories.

## METHODS

### Selection of national newspapers

Ten UK national newspapers were purposively selected to represent the breadth of UK national print media, in terms of newspaper genre (quality, middle-market or tabloid), current political alignment and readership demographics.[30–32] Newspapers included major UK and Scottish titles, and their Sunday equivalents, and comprised: *Telegraph, Times* and *Guardian* (UK quality publications); the *Herald* and the *Scotsman* (Scottish quality publications); the *Daily Mail* and the *Daily Express* (UK middle-market publications); the *Daily Mirror* and the *Daily Star* (UK tabloids) and the *Daily Record* (Scottish tabloid). Full details are presented in online supplementary appendix 1. Scottish titles were considered separately as, in the UK, health is a devolved matter and under the jurisdiction of the Scottish Government, with well-recognised variation in both health strategy and the organisation of service delivery.[33 34]

### Searches

Searches were conducted using the Nexis database; the chosen timeframe was 1 January 2004 until 31 October 2015, which incorporated the implementation of the new General Medical Services (GMS) contract for general practitioners in the UK, with the changes to OOHS provision as previously described.[1 35] All articles containing the terms 'out-of-hours' (≥3 mentions per article) or ('NHS 24' OR 'NHS 111' OR 'NHS Direct' (the names of UK National Health Service (NHS) telephone triage and advice services)) AND 'out-of-hours' (≥1 mention per article) were included. Letters, duplicate news items, opinion pieces without editorial content and articles without a substantial portion of the story (>50% of an article's word count, as judged by researchers) concerning OOHS were excluded. The searches returned 1625 articles in total; we purposively selected all articles published in the years 2005, 2010 and 2015 for detailed content analysis, in order to provide a snapshot of reporting over this timeframe.

### Coding and analysis

A retrospective cross-sectional quantitative content analysis was conducted using a coding pro forma developed iteratively by all the authors.[32 36 37] The pro forma recorded how often, and in which newspapers, stories about OOHS were reported, the type of story (news item, feature article or editorial) and the main theme of the article. Main themes were identified by the study team as those relevant to the research aims, including: OOHS organisation, demand/volume of work, GP contract and personal narrative or case study. Most of the themes were identified by the team prior to coding and were based on a priori knowledge of OOHS provision and research.[38] Further themes were identified inductively as they emerged during the coding process. When a personal narrative was reported, demographic details about the patient, the clinical problem presented to the OOHS and

the outcome were recorded. Articles could be coded as having more than one theme, for example, service organisation and a personal narrative.

HF, SM and CAO read and coded a total of 100 articles using the pro forma. The first 30 articles in each year were triple coded by these authors. All authors discussed this coding and designed the pro forma, adding and refining themes and recorded content as identified (eg, adding themes related to whether articles used supporting statistics or gave advice to readers about accessing OOHS). Following that, the same authors each selected one identified year (2005, 2010 or 2015) and coded the remaining articles from that year. Coding decisions were discussed within the team throughout this process to ensure consistency.

Data were entered into SPSS for statistical analysis. Textual data on the themes contained in the articles were coded numerically (1=yes, theme was reported; 2=no, theme was not reported); thus, the resulting analysis was quantitative.[32] Much of the coding recorded the presence or absence of thematic content, for example, did the article report on the GP contract or not? However, some coding required interpretation of the meaning underlying surface content, described as latent coding,[32 36] such as when recording the tone of an article. Each article's overall tone was coded as being positive, negative or neutral; this was assessed by the researcher coding that paper and discussed with the rest of the research team. Kruskal-Wallis tests were used to test the statistical significance of relationships between article tone and publication, article tone and publication genre and how the median word count of articles varied by publication genre. A $\chi^2$ test was used to test differences in tone between publication genres, while a Wilcoxon signed-rank test

was used to determine whether publications' media tone deviated significantly from a neutral tone. The threshold for statistical significance was set at p<0.01.

### Patient involvement

The stimulus for this study came from work conducted for the Scottish Government's Out-of-Hours Review Group, which included a range of policy, professional and patient stakeholder groups. However, patients were not explicitly involved in the design or interpretation of the work reported here.

### RESULTS

There were 332 articles from the sampled years that met the inclusion criteria: 113 (34.1%) published in 2005, 140 (42.2%) in 2010 and 79 (23.8%) in 2015. Overall, 182 (54.8%) articles were news reports, 44 (13.3%) were features and 99 (29.8%) were editorials. Of the 332 articles, 195 (58.7%) were in the quality press, 99 (29.8%) in middle-market newspapers and 38 (11.5%) in the tabloids. While there was variation across publications and by year, the *Daily Mail/Mail on Sunday* published the most articles (n=76 (22.9% of total)) whereas the *Daily Star/Daily Star Sunday* published just one article (figure 1). The *Guardian/Observer* and *Daily Telegraph/Sunday Telegraph* both published substantially more articles on OOHS in 2010 than in either 2005 or 2015, though it should be noted that the last 2 months of 2015 were not accounted for in the sample.

The overall median word count was 508.0 words (25th to 75th percentiles: 323.0 to 687.5). Median word count was highest in the middle-market newspapers and lowest in the tabloids (Quality: median 491.0 (25th to 75th

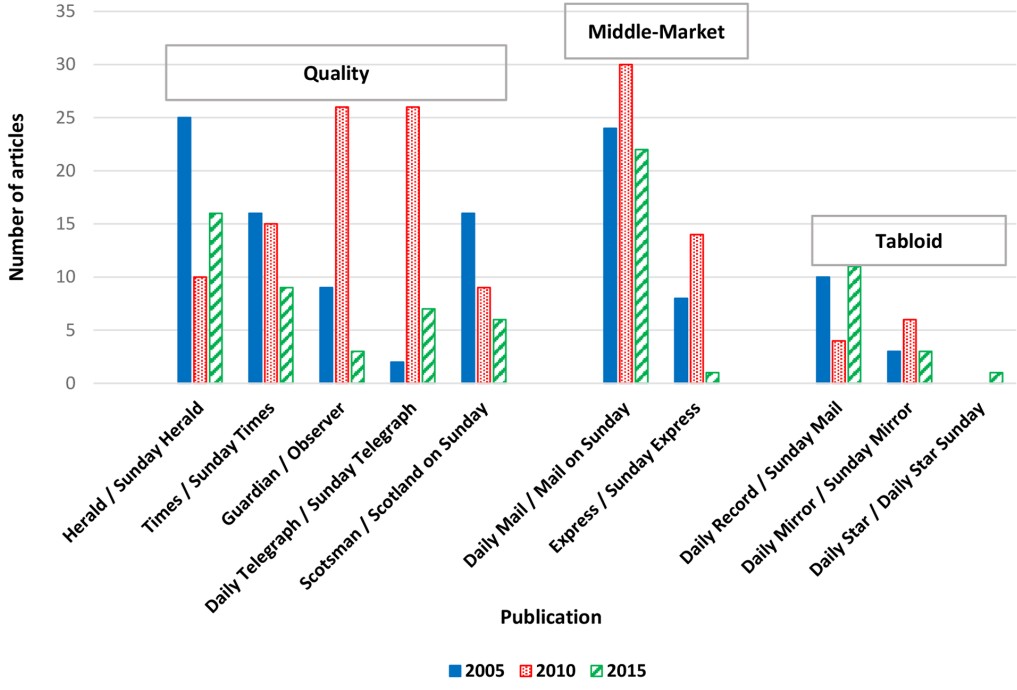

**Figure 1** Article frequency by publication and year.

**Table 1** Reporting of OOHS themes by year and by publication genre (number of times theme coded (%))*

| Theme | Total number of times theme coded (n=621) | Publication year (n, %) | | | Publication genre (n, %) | | |
|---|---|---|---|---|---|---|---|
| | | 2005 (n=174) | 2010 (n=332) | 2015 (n=115) | Quality (n=365) | Middle-market (n=192) | Tabloid (n=64) |
| Service organisation | 139 | 40 (23.0) | 75 (22.6) | 24 (20.9) | 92 (29.5) | 40 (25.8) | 7 (10.9) |
| Case study/Personal narrative | 108 | 13 (7.5) | 83 (25.0) | 12 (10.4) | 62 (19.9) | 33 (21.3) | 13 (20.3) |
| Telephone triage | 95 | 60 (34.3) | 11 (3.3) | 24 (20.9) | 59 (18.9) | 21 (13.5) | 15 (23.4) |
| GP contract | 52 | 18 (10.3) | 30 (9.0) | 4 (3.5) | 25 (8.0) | 25 (16.1) | 2 (3.1) |
| Demand/Volume of work | 44 | 16 (9.2) | 10 (3.0) | 18 (15.7) | 28 (9.0) | 12 (7.7) | 4 (6.2) |
| GP pay | 33 | 12 (6.9) | 17 (5.1) | 4 (3.5) | 15 (4.8) | 13 (8.5) | 5 (7.8) |
| Rurality | 20 | 3 (1.7) | 15 (4.5) | 2 (1.7) | 17 (5.4) | 0 (0.0) | 3 (4.7) |
| Seven day services | 12 | 2 (1.2) | 1 (0.3) | 9 (7.8) | 4 (1.3) | 7 (4.5) | 1 (1.6) |
| Public responsibility/Appropriateness | 10 | 1 (0.6) | 3 (0.9) | 6 (5.2) | 4 (1.3) | 2 (1.3) | 4 (6.2) |
| Cost of services | 8 | 2 (1.2) | 4 (1.3) | 2 (1.7) | 6 (1.9) | 2 (1.3) | 0 (0.0) |
| Other | 100 | 7 (4.0) | 83 (25.0) | 10 (8.7) | 53 (17.0) | 37 (23.9) | 10 (15.7) |

*Articles coded to more than one theme.
GP, general practitioner; OOHS, out-of-hours services.

percentiles: 329.0 to 654.0); middle-market: median 612.0 (25th to 75th percentiles: 445.0 to 804.0; tabloid: median 336.5 (25th to 75th percentiles: 199.3 to 504.0) (Kruskall-Wallis p<0.001).

### Thematic content

Thematic content was coded 621 times across the 332 articles; thus most articles reported more than one theme. The most frequently reported themes were service organisation, personal narratives and telephone triage (table 1), but there were variations by year and publication genre.

Service organisation was the most frequent theme because all other themes or stories within articles were frequently discussed with respect to wider organisational or structural issues. For example, articles focussing on personal narratives would often relate the narrative to staff shortages or OOHS organisational changes. Other subthemes within service organisation included discussion around models of care (eg, commercial agencies' provision of OOHS), IT systems and political discourse on OOHS organisation. Articles reporting personal narratives were second most frequent and were particularly frequent in 2010 (n=83), prompting closer examination of these articles (see below). In 2005, telephone triage was a recurrent theme; 53 of 60 references to telephone triage in 2005 related to a widely publicised report on NHS 24, the then relatively new Scottish telephone triage service, following a fatal accident inquiry.[39] The theme of telephone triage in 2005 also linked to the service organisation theme, because articles reporting telephone triage often discussed wider OOHS organisational issues. Broadly similar frequencies of topics were seen across

the three publication genres, but tabloid newspapers published fewer articles about service organisation.

Articles coded as 'other' encompassed a range of topics, including service response times, patient satisfaction, GP OOH responsibility, confusion around services and quality of care. In each case, there were only one to five articles concerned with each topic. The exception was the theme of 'unsafe non-UK doctors', which was a recurrent theme in 2010, comprising 58 of the 83 articles coded as 'other' in that year. Inspection of these 58 articles found that 53 (91.4%) of these occurrences were in newspaper articles also coded to the theme of 'personal narrative'. This is presented in more detail in the section entitled Reporting of personal narratives.

As well as recording main themes of articles, mentions of specific issues were recorded as part of the pro forma. Twelve (3.6%) OOHS news articles mentioned explicit self-management advice for patients or offered information on how to use OOHS appropriately, and only one article contained advice on accessing OOHS. Conversely, 26 articles (7.8%) described confusion around OOHS. Numerical data focused on OOHS problems (eg, focused on reduced numbers of staff on duty, large number of calls or home visits delayed, high annual costs) were frequently cited in articles but denominator or contextual data were rarely given alongside the headline statistic, for example, the call rate per annum or number of cases of meningitis seen by an OOHS in 1 year.

### Article tone

Overall, 252 of the 332 articles (75.9%) were negative with only 11 (3.3%) positive and 69 (20.1%) neutral

**Table 2** Tone by newspaper publication (Number of articles, (% within newspaper publication))

| Publication | Positive tone (n=11) | Negative tone (n=252) | Neutral tone (n=69) |
|---|---|---|---|
| Quality | | | |
| *Herald/Sunday Herald* (n=51) | 1 (2.0) | 30 (58.8) | 20 (39.2) |
| *Times/Sunday Times* (n=40) | 0 (0.0) | 32 (80.0) | 8 (20.0) |
| *Daily Telegraph/Sunday Telegraph* (n=35) | 1 (2.9) | 26 (74.3) | 8 (22.8) |
| *Guardian/Observer* (n=38) | 2 (5.3) | 29 (76.3) | 7 (18.4) |
| *Scotsman/Scotland on Sunday* (n=31) | 1 (3.2) | 19 (61.3) | 11 (35.5) |
| Middle-Market | | | |
| *Daily Mail/Mail on Sunday* (n=76) | 0 (0.0) | 67 (88.2) | 9 (11.8) |
| *Express/Sunday Express* (n=23) | 0 (0.0) | 19 (82.6) | 4 (17.4) |
| Tabloid | | | |
| *Daily Record/Sunday Mail* (n=25) | 5 (20.0) | 19 (76.0) | 1 (4.0) |
| *Daily Mirror/Sunday Mirror* (n=12) | 1 (8.3) | 10 (83.3) | 1 (8.3) |
| *Daily Star/Daily Star Sunday* (n=1) | 0 (0.0) | 1 (100.0) | 0 (0.0) |

(table 2). Articles published in the middle-market newspapers were more likely to be negative in tone (86/99; 86.9%) compared with the quality newspapers (136/195; 69.7%) or the tabloids (30/38; 79.0%) (Pearson $\chi^2$=35.72, p<0.001). Each individual publication leaned significantly toward a negative tone (Wilcoxon signed-rank p<0.01). While no publication was significantly more positive/negative/neutral than any other (Kruskall-Wallis p=0.025), some—notably the Scottish quality newspapers—tended towards a more neutral tone of reporting (table 2). Tone also varied by year, with 69.9%, 90.7% and 58.2% of articles being negative in tone for years 2005, 2010 and 2015, respectively.

### Reporting of personal narratives

Nearly half of the articles (153; 46.1%) mentioned at least one personal narrative, with the majority portraying such stories as tragedies (132/153; 86.3%). Most featured the personal narrative as the main theme (108/153; 70.6%); however, others commented on a personal narrative in passing, for example, at the end of an article. Where personal narrative was a main theme, the majority described narratives that involved rare diagnoses or problems (104/108; 96.3%). Examples included fatal iatrogenic overdose, fatal sepsis or fatal meningitis in younger patients. Individual cases were also repeated in multiple stories. This was particularly apparent in one story, reported in 2010, of an overseas doctor working in an OOHS who accidentally administered a fatal overdose of diamorphine to a patient during the doctor's first OOHS shift in the UK. Of the 140 articles on OOHS published in 2010, 89 (63.6%) referenced this case. Of these 89 articles, 58 (65.2%) were coded to the theme of 'unsafe non-UK doctors'—meaning that a main theme of these articles was this story. The remaining 31 articles used this story indirectly to illustrate a point in an unrelated article.

### DISCUSSION

This is the first study that describes UK newspaper portrayal of OOHS. Our main finding is that articles on OOHS were significantly more likely to have a negative tone than a positive or neutral tone; middle-market newspapers were significantly more likely to have a negative tone compared with the quality or tabloid newspapers. This negativity was unrelated to newspaper political slant or readership demographics. These findings highlight an over-representation of negative stories around OOHS, as illustrated by the word cloud of the most frequently occurring 'negative' headline words (figure 2). This figure was created by collating all the negative headline terms whereby the size of the word was determined by the frequency that the word appeared in different headlines, with larger words representing more frequently used words. This negative representation of OOHS contrasts with patient views of care, with the 2017 GP patient survey

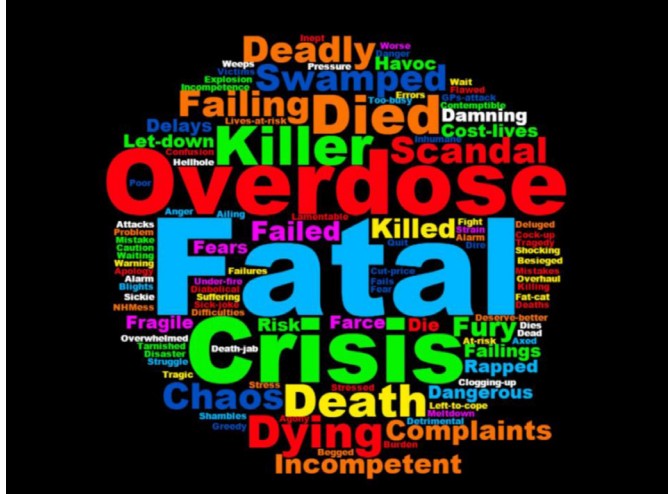

**Figure 2** Word cloud of most frequently appearing negative words in headlines about out-of-hours services services.

in England finding that 66.2% of respondents who had recently accessed OOHS rated their overall experience as 'fairly or very good'.[40] There has, however, been a slight drop in satisfaction since the question was first asked in the 2012 survey, where 70.9% of respondents replied positively.[41] While this slight decrease cannot be attributed to media coverage, it appears part of a pattern of increasingly negative attitudes towards primary care services both in terms of patient satisfaction and in terms of media reporting. Two previous studies of GP portrayal and GP pay identified a change in GP newspaper portrayal over time related to contractual change, with portrayals of both GPs and their salaries becoming more negative after the introduction of the 2004 GMS contract.[28 29] However, our study found much higher levels of negativity, in terms of reporting, than either of the two previous studies. This may have implications for patients' ongoing trust of both OOHS, and GPs in general.

Our results illustrate how a single issue within the OOHS can characterise a large proportion of OOHS-related news. For example, telephone triage services were a dominant theme in 2005, with many articles related to a report published by the Scottish Government on the Scottish telephone triage service, NHS 24.[42] This report was often referred to in the media when reporting on a judicial inquiry into the deaths of a teenager from meningitis and a 30-year-old man from sepsis, both of whom had contacted the triage service prior to their deaths, or in relation to patient complaints of long 'call-back' waiting times for NHS 24. This is consistent with previous media analysis that have highlighted a UK media focus on health service 'crises',[28 29] or on more negative aspects of public health programmes, such as perceived adverse impacts of vaccination.[43 44] This suggests that negative news reports on OOHS are part of a wider and persistent pattern of negative print media reporting about the NHS.

Our analysis also identified the important role that a single case can play in framing media representation of OOHS. Some cases will indeed be worth reporting and a missed diagnosis resulting in a death is always a tragic incident. However, such incidents are rare and persistent or repeated reporting of particular cases could provide a skewed picture of OOHS. For example, the reporting in 2010 of the Coroner's ruling on the unlawful killing of a patient by a doctor working in an OOHS dominated the reporting that year. Although an incomplete calendar year was included for 2015, there appeared to be more OOHS articles published in 2010 than in the other 2 years, and a large proportion of them reported on this case either directly or indirectly. In addition, a higher proportion of articles published in 2010 were negative, compared with the other 2 years sampled. This case was often included in unrelated articles as an exemplar of an OOHS tragedy. This use of 'negative exemplars' was also seen in relation to personal narratives about the presentation of rare clinical problems, such as septicaemia or meningitis, with such cases often used in passing at the end of

unrelated OOHS articles. This is in line with previous media analysis that showed leading US magazines and newspapers over-represented infrequent causes of death while under-representing common causes.[27] Similarly, a study of UK media showed the number of 'deaths-per-news-story' was much higher for common causes (eg, smoking) of death compared with rare causes (eg, vCJD).[43] Public perception of the safety of OOHS could therefore be skewed by an over-representation of rare and tragic cases. This is coupled with low levels of reliable statistical reporting and with very few articles offering advice on when, and how, to access OOHS.

While the media have an important role in holding public bodies and services to account, we believe the media also has a responsibility to portray public services fairly and to provide related impartial information. Misrepresentations may affect people's interaction with, and outcomes from, OOHS either by increasing demand through unnecessary concern over rare illnesses or, alternatively, through eroding public trust and therefore delaying use of the service. Doctors may also be reluctant to work in OOHS where they may be concerned about managing risk in undifferentiated presentations of acute illness and the possible consequences if they are judged to 'make a mistake'.

Despite a preponderance of reporting on crises within and confusion around OOHS services, newspapers offered little practical guidance on accessing care. Thus, to counteract skewed representations of OOHS and their staff, to reduce inappropriate OOHS use and to improve public understanding of health services, we recommend the development of guidelines for media reporting on health services. While the starting point here is OOHS, these could be adapted to other healthcare settings.

### Developing reporting guidelines for OOHS

National and international guidelines already exist for the media reporting of suicides,[45 46] with the threat of failure to adhere to such guidelines in the UK resulting in referral to the UK Press Complaints Commission. Best practice guidelines also exist for UK media reporting of scientific studies.[47] However, these guidelines focus on research findings and omit recommendations specific to health services that are of public health importance. Box 1 contains suggested recommendations on OOHS reporting that could be integrated into current reporting guidelines. Although these guidelines are in response to an analysis of UK print media, they are likely to be relevant in countries with publicly funded healthcare systems and similar media reporting.

### Strengths and limitations

This is the first media analysis of reporting on OOHS services. The analysis covered a comprehensive range of UK print media, including a representative spread of main publication genres, political leaning and readership demographics. The time span covered by

> **Box 1  Media reporting guidelines for articles reporting on OOHS**
>
> 1. Media reporting of OOHS should avoid the practice of adding a 'negative exemplar' at the end of a story unrelated to that personal narrative.
> 2. Media sources are encouraged to consider adopting the practice of 'positive exemplars', namely personal narratives where someone received timely and appropriate care, perhaps for a rare or unusual clinical problem.
> 3. Where personal narratives are reported to depict poor care and specific diseases or illnesses are mentioned, provide further disease-related information or direct readers to further information from providers of impartial health advice. For example, on reporting following a case of fatal sepsis in a young person, direct the reader to the website of a prominent sepsis charity/organisation.
> 4. When reporting on service access problems, direct readers to further information on how and when to navigate services. For example, if reporting on a case of delay in care, direct the reader to NHS Choices or provide the local telephone triage service number.
> 5. When reporting health service-related statistics, provide contextual data. For example, if reporting on how few doctors are covering an OOHS shift, provide information or provide links to information on how many staff normally cover that shift. Or if reporting on costs of a service, direct readers to more information on costs of other public services.
> 6. When reporting on use of an OOHS, provide contextual information such as the number of calls or visits the service received per annum.
> 7. Where possible, provide links to local information or campaigns about alternative sources of healthcare support, for example, local pharmacies.
> 8. OOHS health professionals, managers and service designers should increase, enhance and capitalise on opportunities to work with journalists and editors to help publishers follow these recommendations.

the analysis reflects a time of substantial change to the organisation of OOHS in the UK. The study only included newspaper articles, and there are likely to be other popular news sources excluded from this study that also exert influence on public perceptions, for example, online news and social media. However, while print newspaper circulation is declining, online news is largely dominated by the online counterparts of traditional newspapers, such that analysing online news content may yield similar results. Examining related social media data was beyond the scope of this study but future research in this area would be valuable given the rise in social media use. The combination of deductive and inductive approaches to collect data based on both priori themes and emergent themes, respectively, may affect the reproducibility and therefore generalisability of our findings. Similar analyses in other settings, including international settings, are required. Finally, we cannot rule out individual researcher bias as most articles were coded by a single researcher. However, this bias might be limited as approximately one-third of the articles were independently coded by three researchers after which discrepancies were discussed. Consensus was easily reached before the remaining articles were coded.

## CONCLUSION

In summary, we found UK newspaper OOHS reporting to be generally negative in tone, irrespective of newspaper type, which, in keeping with previous media analyses, included a preponderance of articles describing crises or personal narratives depicting rare and tragic patient stories. Uniquely, we found that media reports related to OOHS can become dominated by a single personal narrative. Our findings provide clear examples of media representations that may negatively affect the public's perceptions of, and interaction with, OOHS. Developing guidelines to encourage responsible reporting on health services may have a role in reducing the risk of skewed public perceptions. Further research that examines public perceptions of OOHS in light of these newspaper representations would develop understanding of the media's role in shaping public opinion of health services.

**Contributors**  HF, SM and CAO conceived the idea and designed the study; HF and CP analysed the data and designed the search strategies, with input from SM and CAO. HF wrote the first draft; CAO led the redrafting and is the guarantor. All authors contributed to the design of the data extraction pro forma and data extraction, contributed to data interpretation, contributed to the final version and agreed to its submission.

**Funding**  This study was unfunded. HF is supported by an NHS Education for Scotland Academic Fellowship. CP's time was funded by the Informing Healthy Public Policy programme (Funded by the Medical Research Council MC_UU_12017–15 and the Chief Scientist Office SPHSU15) of the MRC/CSO Social and Public Health Sciences Unit, University of Glasgow.

**Competing interests**  None declared.

**Patient consent for publication**  Not required.

**Provenance and peer review**  Not commissioned; externally peer reviewed.

**Data sharing statement**  Copies of the search strategy, identified media articles and data extraction pro forma are available on request to HF or COD.

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
