## [Reviewer comments · BMJ Open]

This paper was submitted to a another journal from BMJ but declined for publication following peer review. The authors addressed the reviewers' comments and submitted the revised paper to BMJ Open. The paper was subsequently accepted for publication at BMJ Open.

(This paper received three reviews from its previous journal but only two reviewers agreed to published their review.)

ARTICLE DETAILS

TITLE (PROVISIONAL)	No such thing as bad publicity? A quantitative content analysis of print media representations of primary care out-of-hours services.
AUTHORS	Foster, Hamish; MacDonald, Sara; Patterson, Chris; O'Donnell, Catherine A

VERSION 1 – REVIEW

REVIEWER	Ellen Keizer Institute for Primary Care, Zürich University, Switzerland
REVIEW RETURNED	18-May-2018

GENERAL COMMENTS	In general, I think it is a well-written article on an interesting topic. I think it is important to be aware of the influence of media. I think the discussion of the paper can be improved by making a more extensive comparison with other news. See below for some (minor) comments: Introduction - The introduction is clearly written. The authors are giving a good overview of the situation regarding OOH care. Methods - "HF, SM and CO'D read and coded a total of 100 articles using the pro-forma". Did you all coded the same 100 articles or in total 100? Please clarify. Depending on this, I think it could be a limitation that each researcher coded one year, because you present the results for each year. It would have been better if each researcher coded a mix of the three years. One researcher could code differently than the other (for example more negative). Results - You wrote that the Guardian/Observer and Daily Telegraph/Sunday Telegraph both published significantly more articles on OOHS in 2010 than in either 2005 or 2015. I don't think it is relevant to make this comparison, since 2015 was not included for the whole year. Moreover, this implies that a trend has been studied which, in my opinion, is not possible with only three years.- Since it is a quantitative content analysis, it is logical that numbers are reported. However, I think the article will be more attractive if you write a bit more about the content of the news articles. Maybe, you can give some more
---

	examples (like you did with the example of the unsafe case)  - “Only 12 of OOHS news articles mentioned any explicit self-management advice for patients or offered information on how best to use OOHS” The word ‘only’ sounds a bit ‘subjective’. In the discussion, you could refer to this. Do you think it is a task for newspapers to advice about how to use OOHS? Discussion  - Figure 2 needs more explanation. What have you done? - You refer to a study about patient satisfaction and you wrote that a decrease in patient satisfaction cannot be attributed to media coverage, but is part of a pattern. What do you mean by this? Which pattern? Please, clarify. - To be able to interpret the results, a more extensive comparison with news on other topics is desirable. How does negative reporting on OOHS relate to other news and more specific to news on other healthcare services? Would it be possible that news in general is often negative, and that negative news about OOHS is no exception? I would prefer some reflection about this in the discussion. - Regarding recommendation: would there also be a task for doctors or researchers to put some efforts in reaching the public by news articles on more positive news and education? For example, doctors/researchers could report and explain about OOHS and can also reflect on incidents.
--	---

REVIEWER	Bent Håkan Lindberg Department of General Practice, Antibiotic Center for Primary Care, Institute of Health and Society, University of Oslo, Norway.
REVIEW RETURNED	23-Jul-2018

GENERAL COMMENTS	The authors have written an interesting paper about portrayals of primary care out-of-hours services in print media. The article shows how newspapers tend to report negatively about this crucial health service. I have a few questions and remarks for the authors: Page 2, line 32-33 and page 7, line 49-52; What does >50 % of the story mean? How has this been assessed? Page 5, Line 36-37 with reference number 7; I read this reference differently, finding that it talks about inappropriate use of hospital EDs due to malfunctioning OOHS. I suggest that you delete “and a concern about the supposed inappropriate use of OOHS”, including reference 7. Page 9, line 5-6; I am not sure that I understand what you mean with “variation in median article word count and publication genre”. What kind of word count? Could you explain better? Page 9, line 33-34; You write “there were 332 articles”. I suggest you add “meeting the inclusion criteria” or something similar to clarify. Page 9, line 43-44; I am not sure about the expression “the most articles”. I suggest you rephrase. Page 10, line 25; Does the fatal accident enquiry story have a reference? Page 11, line 5-6; I suggest that “how best to use” is changed to “appropriate use” Page 11, line 39-41; Would the phrase sound better if you write “.....of articles being negative in tone.....”? Page 12, line 56; I don’t find that any of the two references (26 and
---

	27) explain the 2004 GMS contract, but I don't have access to the full text articles. Page 13, line 11-18; I suggest that you delete the phrase "It is important to acknowledge.....rare occurrences" and the phrase "Our argument.....picture of OOHS". I think this will make the discussion more focused. Page 13, line 52-55; Why do you think the media has a responsibility to portray fair representations of public services? This should be followed by an argument or by a reference. Page 15, line 49 to page 16, line 5; I think there are several good reasons not to include social media in your analyses, but you have only argued for not including online newspapers. Box 1: First point: This is not a law. I therefore suggest that you change "Media reporting of OOHS must stop the practice" to "Media reporting of OOHS should avoid the practice of"
--	--

REVIEWER	Sanna Koskinen, PhD, Postdoctoral Researcher University of Turku, Department of Nursing Science, Finland
REVIEW RETURNED	22-Oct-2018

GENERAL COMMENTS	Thank you for the opportunity to review this interesting manuscript. It provides a less-used, although not unique, perspective on the healthcare services provided in one country. The manuscript has the potential to be published. However, there are a number of issues the authors should consider. Title:  • The title of the manuscript is quite long. Therefore, I suggest shortening it by removing the interrogative clause. This clause does not illustrate the findings nor does it present a unique thought. • No portrayals are presented in the findings. Therefore, I would also remove "portrayals". • A good title could be "Primary care out-of-hours services in print media: A quantitative content analysis". Abstract  • In the design section, an analysis method is described. I recommend adding the used design. • Moreover, I suggest revising the last sentence of the conclusion section based on my comments presented later concerning the manuscript. Strengths and Limitations  • These bullet points relate only to the coverage (breadth) of the included material in terms of time and sources. I suggest providing some other aspects as well (please see my comments presented later concerning the manuscript). Introduction  • I suggest clarifying the main difference(s) between the emergency departments and the primary care OOHS for a reader outside the UK. I recommend giving a short description about this. Methods  • Like in the abstract, no design is reported. I recommend adding the used design. Searches  • It is reported that the chosen timeframe began on 1 January
--

2004. Everywhere else, it is reported that the examined years were 2005, 2010 and 2015. Is 2004 a spelling mistake, or were there no articles in 2004?

- About the exclusion criteria: Firstly, what is meant by “duplicate news items”? Do you mean that the search from Nexis produced duplicate references? I assume there could be several news (articles) about the same thing (e.g. on different days). However, this is not duplicate news. Secondly, how was the exclusion criterion “substantial portion of the story (>50%) concerning OOHS” determined? In other words, was it based on the number of words in the article or something else? Thirdly, how many of the authors were involved in the selection of the articles?

Coding and analysis

- The description of the analysis process needs to be elaborated upon. Firstly, was the reasoning approach deductive, inductive or abductive? Secondly, what were the selected content areas – OOHS organisation, demand/volume of work, GP contract and personal narrative or case study – based on? Was there any standard classification or objective evidence to support the selection of these areas in advance?

Patient involvement

- The information presented here is relevant, but could it be integrated into the manuscript somewhere else, perhaps the introduction, to keep the text flowing?

Thematic content

- Overall, the reporting is quite fragmented and contains detailed results that can be read in Table 1. Instead, I would like more elaboration upon the most frequently reported themes and whether there was some change in the years 2005, 2010 and 2015 concerning each team. Further, it would be good to check the logic of the writing. For example, three of the most frequently reported themes are mentioned, and then the text explains the third theme. Please explain more about them in the same order as mentioned.

Discussion

- If you want to use a word cloud in your manuscript, its place is in the results, not in the discussion. In addition, it would be good to elaborate the process of how the word cloud was created. Why were only negative words from headlines included? Why not all words?
- The discussion somewhat repeats the results and does not go beyond the perspectives presented in the introduction. It would be beneficial to have international perspectives on the issues, not just a UK perspective. In addition, I would like to read about what healthcare professionals should and could do to promote a more positive image of certain health services.
- In my opinion, a scientific publication is not the most suitable forum to present reporting guidelines for the media. Surely, an effort should be made to implement these results or to at least raise a public discussion about the consequences of unfavourable media coverage. Still, I am not in favour of proposing recommendations here.

Strengths and limitations

- As mentioned earlier, the presented limitations relate only to the coverage (breadth) of the included material in terms of time and sources. I think several other perspectives should be considered.

	For instance, was the deductive approach suitable, and how reliable was the analysis? Conclusion  • Overall, it would be beneficial to rewrite the conclusions. • As said in the manuscript, this is the first media analysis of reporting on OOHS services. Therefore, I recommend toning down expressions like “findings provide clear examples of unhelpful media misrepresentation”. In addition, the following sentences are not conclusions and should be removed: “OOHS articles frequently described personal narratives depicting rare and tragic patient stories, even when the main story was unrelated to the personal narrative. In 2010, reporting frequently focused on unsafe non-UK doctors’ due to a single personal narrative.”
--	--

VERSION 1 – AUTHOR RESPONSE

Reviewer comment		Authors’ response
Reviewer 1		
1.0	In general, I think it is a well-written article on an interesting topic. I think it is important to be aware of the influence of media. I think the discussion of the paper can be improved by making a more extensive comparison with other news. See below for some (minor) comments:	We thank the reviewer for their positive comments. We have tried to respond their comments or questions line by line below.
1.1	Introduction - The introduction is clearly written. The authors are giving a good overview of the situation regarding OOH care.	We thank the reviewer for this positive comment.
1.2	Methods - “HF, SM and CO’D read and coded a total of 100 articles using the pro-forma”. Did you all coded the same 100 articles or in total 100? Please clarify. Depending on this, I think it could be a limitation that each researcher coded one year, because you present the results for each year. It would have been better if each researcher coded a mix of the three years. One researcher could code differently than the other (for example more negative).	We thank the reviewer for pointing out that this aspect of the method was not clear and on the potential limitation. We hope our method is now fully explained and we have adjusted the manuscript to read (from Page 8, line 140): HF, SM and CO’D read and coded a total of 100 articles using the pro-forma. The first 30 articles in each year were triple coded by these authors. All authors discussed this coding and designed the pro-forma, adding and refining themes and recorded content as identified (e.g. adding themes related to whether articles used supporting statistics or gave advice to readers about accessing OOHS). Following that, the same authors each selected one identified year (2005, 2010 or 2015) and coded the remaining articles from that year. Coding decisions were

		discussed within the team throughout this process to ensure consistency. And also in the strengths and limitations section (Page 17, line 326): Finally, we cannot rule out individual researcher bias as most articles were coded by a single researcher. However, this bias might be limited as approximately one third of the articles were independently coded by three researchers after which discrepancies were discussed. Consensus was easily reached before the remaining articles were coded.
Results		
1.3	You wrote that the Guardian/ Observer and Daily Telegraph/ Sunday Telegraph both published significantly more articles on OOHS in 2010 than in either 2005 or 2015. I don't think it is relevant to make this comparison, since 2015 was not included for the whole year. Moreover, this implies that a trend has been studied which, in my opinion, is not possible with only three years.	We thank the reviewer for this comment and agree that comparisons between years are not like-for-like because 2015 articles were collected from an incomplete calendar year. Therefore we have changed the following sentence in the results section (Page 10, from line 173): The Guardian/Observer and Daily Telegraph/Sunday Telegraph both published substantially more articles on OOHS in 2010 than in either 2005 or 2015, though it should be noted that the last two months of 2015 were not accounted for in the sample. In addition, within the discussion section we have altered page 14 from line 274 to read: Although an incomplete calendar year was included for 2015, there appeared to be more OOHS articles published in 2010 than in the other two years, and a large proportion of them reported on this case either directly or indirectly
1.4	- Since it is a quantitative content analysis, it is logical that numbers are	We thank the reviewer for this comment and agree that examples from the newspaper

	reported. However, I think the article will be more attractive if you write a bit more about the content of the news articles. Maybe, you can give some more examples (like you did with the example of the unsafe case)	articles can help make the article more attractive. We have now illustrated some of the general content of the articles in the ‘thematic content’ section of the results and have added to this (see lines 185 to 197). We have also added to the discussion section to include more exemplar content on other areas where a single issue dominated reporting. The addition to the discussion section now reads: Our results illustrate how a single issue within the OOHS can characterise a large proportion of OOHS related news. For example, telephone triage services were a dominant theme in 2005, with many articles related to a report published by the Scottish Government on the Scottish telephone triage service, NHS 24.³⁷ This report was often referred to in the media when reporting on a judicial inquiry into the deaths of a teenager from meningitis and a 30 year old man from sepsis, both of whom had contacted the triage service prior to their deaths, or in relation to patient complaints of long ‘call-back’ waiting times for NHS24. This is consistent with previous media analysis that have highlighted a UK media focus on health service ‘crises’, [28 29] or on more negative aspects of public health programmes, such as perceived adverse impacts of vaccination. [43 44] This suggests that negative news reports on OOHS are part of a wider and persistent pattern of negative print media reporting about the NHS. We have not added more detailed information on content beyond these two examples as there was a clear preponderance of articles focussing on these issues. Inclusion of further content is beyond the remit of our methods and would require full qualitative analysis.
1.5	- “Only 12 of OOHS news articles mentioned any explicit self-management advice for patients or offered information on how best to use OOHS” The word ‘only’ sounds a bit ‘subjective’. In the discussion, you	We thank the reviewer for this point and we agree that the word ‘only’ implies a lack and therefore a judgement. We have removed the word ‘only’ from the sentence.

	could refer to this. Do you think it is a task for newspapers to advice about how to use OOHS?	We have made clearer reference to this in the discussion (Page 15, line 285): Public perception of the safety of OOHS could therefore be skewed by an over-representation of rare and tragic cases. This is coupled with low levels of reliable statistical reporting and with very few articles offering advice on when and how to access OOHS. We feel that there is a role for newspapers to direct readers to reliable sources of information that advises how best to use and access OOHS, a public service. We suggest this is especially important when articles focus on an example of an OOHS access problem (e.g. a fatality linked to delayed access). As a result, we have included this in our recommendations for media reporting guidelines for articles reporting on OOHS (Box 1, point 4).
Discussion		
1.6	- Figure 2 needs more explanation. What have you done?	We have added in the following sentence (page 13, line 244): This figure was created by collating all the negative headline terms whereby the size of the word was determined by the frequency that the word appeared in different headlines, with larger words representing more frequently used words. We appreciate that this methodology for creating a word cloud has many limitations not least the fact that we took negative sounding terms from headlines that themselves may not have been negative in tone thereby skewing the appearance of the word cloud to being overly negative. For this reason we did not include this in the formal methods section. However, this approach provides an image that uses words taken from the headlines of included articles and illustrates our main result of the negative tone of OOHS articles.
1.7	- You refer to a study about patient satisfaction and you wrote that a decrease in patient satisfaction cannot	We thank the reviewer for this comment and have attempted to clarify this sentence.

	be attributed to media coverage, but is part of a pattern. What do you mean by this? Which pattern? Please, clarify.	We have altered the sentence to read (page 13, line 50): While this slight decrease cannot be attributed to media coverage, it appears part of a pattern of increasingly negative attitudes towards primary care services both in terms of patient satisfaction and in terms of media reporting.
1.8	- To be able to interpret the results, a more extensive comparison with news on other topics is desirable. How does negative reporting on OOHS relate to other news and more specific to news on other healthcare services? Would it be possible that news in general is often negative, and that negative news about OOHS is no exception? I would prefer some reflection about this in the discussion.	We thank the reviewer for this valid point and we have included more information on this including a new reference in the discussion (Page 14, line 264): This is consistent with previous media analysis that have highlighted a UK media focus on health service 'crises',[28 29] or on more negative aspects of public health programmes, such as perceived adverse impacts of vaccination.[43 44] This suggests that negative news reports on OOHS are part of a wider and persistent pattern of media reporting about the NHS.
1.9	- Regarding recommendation: would there also be a task for doctors or researchers to put some efforts in reaching the public by news articles on more positive news and education? For example, doctors/ researchers could report and explain about OOHS and can also reflect on incidents.	We thank the reviewer for this valid comment and we agree researchers and health care practitioners also have a responsibility to contribute to raising awareness around OOHS and responsible use of health services. There are already numerous ways in which OOHS providers and designers in the UK have educational and awareness raising roles. However, we believe the primary responsibility for fair and accurate news reporting lies with the publishing organisations themselves. Nevertheless, we would welcome new and improved opportunities for academics and health practitioners to work with media outlets and increase their ability to follow our recommendations (Box 1). We agree that more opportunities for OOHS professionals to reflect on OOHS incidents in news reports would be beneficial and although these opportunities might ultimately be controlled by news publishers we have added another recommendation (see Box 1, Point 8) to encourage these opportunities. 8. OOHS health professionals, managers and service designers should increase, enhance and capitalise on opportunities to work with journalists and editors in order to help publishers follow these recommendations.

		As an aside, we believe that if all of our recommendations are followed this would result in improved communication and links between health service providers, researchers and media organisations. For example, to report recommended contextual health service related statistics, journalists or editors would likely need to communicate with health service providers.
Reviewer 2		
2.0	The authors have written an interesting paper about portrayals of primary care out-of-hours services in print media. The article shows how newspapers tend to report negatively about this crucial health service. I have a few questions and remarks for the authors:	We thank the reviewer for their helpful comments.
2.1	Page 2, line 32-33 and page 7, line 49-52; What does >50 % of the story mean? How has this been assessed?	We thank the reviewer for raising this point. Greater than 50% of the story refers to the word count. This was not assessed formally as articles tended to be overwhelmingly about the topic or overwhelmingly about something else. Therefore this was a judgement made on part of the researcher; it is, however, a standard approach for this type of content analysis. We have altered the text to read (page 2, line 31 and page 7, line 123): Letters, duplicate news items, opinion pieces without editorial content and articles without a substantial portion of the story (>50% of an article's word count, as judged by researchers) concerning OOHS were excluded.
2.2	Page 5, Line 36-37 with reference number 7; I read this reference differently, finding that it talks about inappropriate use of hospital EDs due to malfunctioning OOHS. I suggest that you delete "and a concern about the supposed inappropriate use of OOHS", including reference 7.	We thank the reviewer for this comment and agree that reference 7 talks about inappropriate ED visits related to OOHS organisation. However, we feel that there is often a concern around inappropriate OOHS use as well as inappropriate ED use. As a result we have used different references to evidence that. Page 5, from line 76 now reads: A second driver for change is increasing patient demand, due in part to ageing populations and associated multimorbidity,[12 13] and a concern about the supposed 'inappropriate' use of OOHS.[14 15]

2.3	Page 9, line 5-6; I am not sure that I understand what you mean with “variation in median article word count and publication genre”. What kind of word count? Could you explain better?	In order to make this clearer we have reworded the sentence to read (page 9, line 153): Kruskal-Wallis tests were used to test the statistical significance of relationships between article tone and publication; article tone and publication genre; and how the median word count of articles varied by publication genre.
2.4	Page 9, line 33-34; You write “there were 332 articles”. I suggest you add “meeting the inclusion criteria” or something similar to clarify.	We thank the reviewer for this suggestion which agree can add clarity. Page 9 line 166 now reads: There were 332 articles from the sampled years that met the inclusion criteria:
2.5	Page 9, line 43-44; I am not sure about the expression “the most articles”. I suggest you rephrase.	We thank the reviewer for this comment but we feel the original sentence clearly states how the Daily Mail publication published more OOHS related articles than any other publication. We have therefore not edited this sentence (page 9, line 170): While there was variation across publications and by year, the Daily Mail/Mail on Sunday published the most articles...
2.6	Page 10, line 25; Does the fatal accident enquiry story have a reference?	We have added a reference for this (page 10, line 193)
2.7	Page 11, line 5-6; I suggest that “how best to use” is changed to “appropriate use”	We thank the reviewer for this suggestion we have added the word ‘appropriately’ to this sentence, which now reads (page 11, line 205): Twelve (3.6%) OOHS news articles mentioned explicit self-management advice for patients or offered information on how to use OOHS appropriately, and only 1 article contained advice on accessing OOHS.
2.8	Page 11, line 39-41; Would the phrase sound better if you write “.....of articles being negative in tone.....”?	Thank you for this we have changed the sentence to include the word ‘being’ (Page 12, line 220). Tone also varied by year, with 69.9%, 90.7% and 58.2% of articles being negative in tone for years 2005, 2010, and 2015 respectively.
2.9	Page 12, line 56; I don’t find that any of the two references (26 and 27) explain the 2004 GMS contract, but I don’t have access to the full text articles.	We thank the reviewer for highlighting this issue. However these references were not references to explain the 2004 GMS contract but rather references for evidence to show more

		negative newspaper portrayal of UK GPs over time. In response we have now referenced the 2004 GMS contract in the methods section where we first mention this specific policy change and included a new reference here that explained more about the contract over and above the changes to OOHS (Refs 1 & 35 on page 7, line 120).
2.10	Page 13, line 11-18; I suggest that you delete the phrase “It is important to acknowledge.....rare occurrences” and the phrase “Our argument.....picture of OOHS”. I think this will make the discussion more focused.	We thank the reviewer for their suggestion on how to make the discussion more focussed. However, we feel that this section of this discussion is highlighting one of our main findings – that there is a preponderance of reporting of rare and tragic cases. However, at the same time, we want to acknowledge the media’s rightful role in reporting cases of missed diagnoses etc. that might be part of the media’s role in holding public services to account. We have trimmed these sentences to remain concise. We hope this is an adequate response. The respective sentences now read (page 14, line 269): Some cases will indeed be worth reporting and a missed diagnosis resulting in a death is always a tragic incident. However, such incidents are rare and persistent or repeated reporting of particular cases could provide a skewed picture of OOHS.
2.11	Page 13, line 52-55; Why do you think the media has a responsibility to portray fair representations of public services? This should be followed by an argument or by a reference.	We thank the reviewer for this valid point. There is no formal responsibility for media publications to portray fair representations. However, as per guidelines on reporting of scientific studies or reporting on suicides we believe the media is de facto in a position of responsibility to report on public services in a way that does not increase the risk of OOHS misrepresentations and subsequent inappropriate use. We have changed the sentence to add ‘we believe’ and the subsequent sentence forms our argument for why we believe that to be the case. We have no reference for these statements. The sentences now read (from page 15, line 287):

		While the media have an important role in holding public bodies and services to account, we believe the media also has a responsibility to portray public services fairly and to provide related impartial information. Misrepresentations may affect people's interaction with, and outcomes from, OOHS either by increasing demand through unnecessary concern over rare illnesses or, alternatively, through eroding public trust and therefore delaying use of the service.
2.12	Page 15, line 49 to page 16, line 5; I think there are several good reasons not to include social media in your analyses, but you have only argued for not including online newspapers.	We thank the reviewer for this point and agree that we have not stated why we did not look at social media data; that it was beyond the scope of the project. We think examining social media could be a valuable thing to do and have added the following to the strengths and limitation section (page 17, line 320): Examining related social media data was beyond the scope of this study but future research in this area would be valuable given the rise in social media use.
2.13	Box 1: First point: This is not a law. I therefore suggest that you change "Media reporting of OOHS must stop the practice" to "Media reporting of OOHS should avoid the practice of"	We thank the reviewer for this point. We have changed the first point in Box 1 to say 'should avoid'.
Reviewer 3		
3.0	Thank you for the opportunity to review this interesting manuscript. It provides a less-used, although not unique, perspective on the healthcare services provided in one country. The manuscript has the potential to be published. However, there are a number of issues the authors should consider.	We thank the reviewer for their positive comments.
3.1	Title:  • The title of the manuscript is quite long. Therefore, I suggest shortening it by removing the interrogative clause. This clause does not illustrate the findings nor does it present a unique thought. 	We thank the reviewer for their suggestion regarding shortening the title. However, we feel that we wanted to have a title that might catch a reader's eye and feel the 'interrogative clause' raises the question that negative representations of OOHS might have negative impacts on OOHS or patient outcomes. We have shortened the title slightly and we are very happy to go with an editorial decision on the title length.

	 • No portrayals are presented in the findings. Therefore, I would also remove “portrayals”. • A good title could be “Primary care out-of-hours services in print media: A quantitative content analysis”. 	We thank the reviewer for their suggestion. We have used ‘representations’ instead to maintain consistency within the manuscript. We thank there reviewer for their recommendation and we are happy to be guided by the editorial team on the title. Currently the title reads: No such thing as bad publicity? A quantitative content analysis of print media representations of primary care out-of-hours services.
3.2	Abstract  • In the design section, an analysis method is described. I recommend adding the used design. • Moreover, I suggest revising the last sentence of the conclusion section based on my comments presented later concerning the manuscript. 	We thank the reviewer for their recommendation we have added some description to help clarify the design (page 2, line 24): A retrospective cross sectional quantitative content analysis of all articles published in 2005, 2010 and 2015. Please see our response to your later comments regarding the conclusion (3.12) and please see our change to the abstract conclusion (page 3, lines 45-48): In this first reported analysis of UK national newspaper coverage of OOHS, media representation is generally negative in tone, with frequent reports of ‘negative exemplars’ of OOHS crises and fatal individual patient cases with little or no contextualisation. We present recommendations for the future reporting of OOHS, which could apply to the reporting of healthcare services more generally.
3.3	Strengths and Limitations  • These bullet points relate only to the coverage (breadth) of the included material in terms of time and sources. I suggest providing some other aspects as well (please see my comments presented later concerning the manuscript). 	We thank the reviewer for this point. Please see our response to your further point below 3.11 where we show how we have added more to the strengths and limitations. Our strengths and limitations bullet points are organised according to the criteria of BMJ Open, however we have added one more bullet point (page 4, line 58): Analysis method may limit generalisability and researcher bias could not be eliminated.
3.4	Introduction  • I suggest clarifying the main difference(s) between the emergency departments and the primary care OOHS for a reader outside the UK. I 	We thank the reviewer for this point. We have added a sentence describing EDs so the introduction now reads (Page 5, line 60 onwards): Primary care out-of-hours services (OOHS) is defined as primary care provided when family

	recommend giving a short description about this.	doctors' surgeries are closed (weekdays from 1800hr to 0800hr, weekends and public holidays) and is often delivered in community-based clinic settings. Such services generally provide care for conditions that are not life-threatening, in contrast to care delivered by hospital-based emergency departments (ED), which is available at all times and designed to manage more urgent and life-threatening problems. OOHS have seen extensive policy and structural change in many health care systems.¹⁻⁴
3.5	Methods  • Like in the abstract, no design is reported. I recommend adding the used design. 	We have added 'retrospective cross-sectional' into the methods to further describe the design of the content analysis (page 8, line 128).
3.6	Searches  • It is reported that the chosen timeframe began on 1 January 2004. Everywhere else, it is reported that the examined years were 2005, 2010 and 2015. Is 2004 a spelling mistake, or were there no articles in 2004? • About the exclusion criteria: Firstly, what is meant by "duplicate news items"? Do you mean that the search from Nexis produced duplicate references? I assume there could be several news (articles) about the same thing (e.g. on different days). However, this is not duplicate news. Secondly, how was the exclusion criterion "substantial portion of the story (>50%) concerning OOHS" determined? In other words, was it based on the number of words in the article or something else? Thirdly, how many of the authors were involved in the selection of the articles? 	We thank the reviewer for their query. 2004 is not a typo, as that was the first year we included in our search as we wanted to start from the new 2004 GMS contract. However, after retrieving a large number of articles we purposively selected years 2005, 2010, 2015 for analysis to: 1) limit the number of articles for analysis, 2) include 2015 which, at the time, included the latest articles available; but 3) provide a snap shot of content across 3 equally spaced years. We thank the reviewer for questioning 'duplicate news items'. 'Duplicates' refers to multiple versions of the same article from different editions, where only one version of that article was retained in the sample. They were duplicate news items. In relation to how the exclusion criterion of a "substantial portion of the story (>50%) concerning OOHS" was determined please see our response to reviewer comment 2.1 where we address the same question. In response to the final question in this section, all authors were involved in deciding on the selected years as this was team consensus. The decision to include or exclude an article was done during the coding process. In response to another reviewer we have altered the methods section to now read: (Page 8, line 139)

		HF, SM and CO'D read and coded a total of 100 articles (using the pro-forma. The first 30 articles in each year were triple coded by these authors. All authors discussed this coding and designed the pro-forma, adding and refining themes and recorded content as identified (e.g. adding themes related to whether articles used supporting statistics or gave advice to readers about accessing OOHS). Following that, the same authors each selected one identified year (2005, 2010 or 2015) and coded the remaining articles from that year. Coding decisions were discussed within the team throughout this process to ensure consistency
3.7	Coding and analysis  The description of the analysis process needs to be elaborated upon. Firstly, was the reasoning approach deductive, inductive or abductive? Secondly, what were the selected content areas – OOHS organisation, demand/volume of work, GP contract and personal narrative or case study – based on? Was there any standard classification or objective evidence to support the selection of these areas in advance? 	We thank the reviewer for these questions regarding the analysis. We did not use an established framework for coding but rather developed one based on a priori knowledge of this area of service provision and developed the coding pro forma iteratively as themes emerged. This hopefully answers the second question as these themes were frequently based on previous knowledge of this area of service provision. There is no standard or objective evidence to support the selection of these areas in advance. We have added the following to the methods section to help clarify this: (page 8, line 133) Most of the themes were identified by the team prior to coding and were based on a priori knowledge of OOHS provision and research.[38] Further themes were identified inductively as they emerged during the coding process.
3.8	Patient involvement  The information presented here is relevant, but could it be integrated into the manuscript somewhere else, perhaps the introduction, to keep the text flowing? 	We thank the reviewer for this point. This section is here as this is recommended in the BMJ Open authors information section: https://bmjopen.bmj.com/pages/authors/ “To support co-production of research we request that authors provide a Patient and Public Involvement statement in the methods section of their papers.”
3.9	Thematic content  Overall, the reporting is quite fragmented and contains detailed results that can be read in Table 1. Instead, I would like more elaboration 	We thank the reviewer for this comment. We have rearranged the thematic content to follow the order of frequency from Table 1, discussing the three most frequent themes. We have also

	upon the most frequently reported themes and whether there was some change in the years 2005, 2010 and 2015 concerning each team. Further, it would be good to check the logic of the writing. For example, three of the most frequently reported themes are mentioned, and then the text explains the third theme. Please explain more about them in the same order as mentioned.	added additional information regarding the 'service organisation' theme (page 10, line 184).
3.10	Discussion  • If you want to use a word cloud in your manuscript, its place is in the results, not in the discussion. In addition, it would be good to elaborate the process of how the word cloud was created. Why were only negative words from headlines included? Why not all words? • The discussion somewhat repeats the results and does not go beyond the perspectives presented in the introduction. It would be beneficial to have international perspectives on the issues, not just a UK perspective. In addition, I would like to read about what healthcare professionals should and could do to promote a more positive image of certain health services. • In my opinion, a scientific publication is not the most suitable forum to present reporting guidelines for the media. Surely, an effort should be made to implement these results or to at least raise a public discussion about the consequences of unfavourable media coverage. Still, I am not in favour of proposing recommendations here. 	We thank the reviewer for raising this point. Please see our response to reviewer comment 1.6 for more detail but we have changed the section in the discussion to read (page 13, line 243): This figure was created by collating all the negative headline terms whereby the size of the word was determined by the frequency that the word appeared in different headlines, with larger words representing more frequently used words. We thank the reviewer for this comment. In the introduction we describe OOHS, their recent changes, purported drivers for change, and then discuss the potential influence of media on health issues. We finish by highlighting that, to our knowledge, there has been no previous examination of how UK or international newspapers portray OOHS (page 6, line 97). In the discussion we follow the guidance of the journal and reiterate our main results, then compare it to survey results of patients' experience of OOHS and to previous similar research that examined GP and GP portrayal in newspapers. We raise a potential implication of an erosion of trust in OOHS. Please see our addition to the discussion that describes more content from the newspapers in relation to one of the most frequent themes and relates this to previous media analysis (page 13, lines 257-267). We then highlight the impact that a preponderance of reporting of a single, rare and fatal personal narrative might have on both public perception of OOHS and doctors' willingness to work in OOHS settings. We believe that the discussion till this point provides

		the reasoning for developing media reporting guidelines for OOHS. We agree that an international perspective is important even though these guidelines are in response to analysis of UK newspapers only. We have added to the conclusion (page 16, line 307): Although these guidelines are in response to an analysis of UK print media they are likely to be relevant in countries with publicly funded healthcare systems and comparable media reporting. In relation to what healthcare professionals' roles please see our response to reviewer comment 1.9 above where we describe an additional recommendation on this point. While we respect the reviewer's opinion on whether recommendations on media reporting guidelines in a scientific publication is debatable, we disagree that these should be removed. We believe that not only is it ethical to make recommendations that try to redress inaccuracies or potentially harmful representations but it is also critical to use the peer reviewed publication forum to contribute to public health and policy debates. We believe that the incorporation of recommendations builds on the work of Weishaar et al who recently argued that we need to understand better how to frame public health messages and strategies to counter the media framing that undermines public health goals (BMC Public Health 2016; 16: 899).
3.11	Strengths and limitations  • As mentioned earlier, the presented limitations relate only to the coverage (breadth) of the included material in terms of time and sources. I think several other perspectives should be considered. For instance, was the deductive approach suitable, and how reliable was the analysis? 	We appreciate the reviewer's suggestion here. We have made a substantial addition to the strengths and limitations section (page 17, from line 320): Examining related social media data was beyond the scope of this study but future research in this area would be valuable given the rise in social media use. The combination of deductive and inductive approaches to collect data based on both priori themes and emergent themes respectively may affect the reproducibility and therefore generalisability of our findings. Similar analyses in other settings, including international settings, are required.

		Finally, we cannot rule out individual researcher bias as most articles were coded by a single researcher. However, this bias might be limited as approximately one third of the articles were independently coded by three researchers after which discrepancies were discussed. Consensus was easily reached before the remaining articles were coded.
3.12	Conclusion  • Overall, it would be beneficial to rewrite the conclusions. • As said in the manuscript, this is the first media analysis of reporting on OOHS services. Therefore, I recommend toning down expressions like “findings provide clear examples of unhelpful media misrepresentation”. In addition, the following sentences are not conclusions and should be removed: “OOHS articles frequently described personal narratives depicting rare and tragic patient stories, even when the main story was unrelated to the personal narrative. In 2010, reporting frequently focused on unsafe non-UK doctors’ due to a single personal narrative.” 	We thank the reviewer for their suggestion. We have made the following changes to the conclusion (page 17, line 332 onwards): In summary, we found UK newspaper reports on OOHS to be generally negative in tone, irrespective of newspaper type, and, in keeping with previous media analyses, a preponderance of articles describing crises or personal narratives depicting rare and tragic patient stories. Uniquely, we found that media reports related to OOHS can become dominated by a single personal narrative. Our findings provide clear examples of media representations that may negatively affect the public’s perceptions of, and interaction with, OOHS. Developing guidelines to encourage responsible reporting on health services may have a role in reducing the risk of skewed public perceptions. Further research that examines public perceptions of OOHS in light of these newspaper representations would develop understanding of the media’s role in shaping public opinion of health services.

VERSION 2 – REVIEW

REVIEWER	Bent Håkan Lindberg The Antibiotic Centre for Primary Care, Department of General Practice, Institute of Health and Society, University of Oslo
REVIEW RETURNED	03-Dec-2018
GENERAL COMMENTS	The article is well written and interesting. After your revision I have no further comments or suggestions.